# Cellular Uptake and Tissue Retention of Microplastics in Black Soldier Fly Larvae

**DOI:** 10.3390/insects16111169

**Published:** 2025-11-16

**Authors:** Claudiu-Nicusor Ionica, Romelia Pop, Dragos Hodor, Irina Constantin, Ana Hiruta, Alexia-Teodora Hota, Alexandru Flaviu Tabaran, Sorana Daina, Andrei-Radu Szakacs, Adrian Macri

**Affiliations:** 1Department of Animal Nutrition, Faculty of Veterinary Medicine, University of Agricultural Sciences and Veterinary Medicine of Cluj-Napoca, Calea Manaștur, 400372 Cluj-Napoca, Romania; claudiu-nicusor.ionica@usamvcluj.ro (C.-N.I.); sorana.matei@usamvcluj.ro (S.D.); andrei.szakacs@usamvcluj.ro (A.-R.S.); adrian.macri@usamvcluj.ro (A.M.); 2Department of Pathology, Faculty of Veterinary Medicine, University of Agricultural Sciences and Veterinary Medicine of Cluj-Napoca, Calea Manaștur, 400372 Cluj-Napoca, Romania; dragos.hodor@usamvcluj.ro (D.H.); irina.constantin@usamvcluj.ro (I.C.); ana.hiruta@usamvcluj.ro (A.H.); alexia-teodora.hota@usamvcluj.ro (A.-T.H.); alexandru.tabaran@usamvcluj.ro (A.F.T.)

**Keywords:** confocal microscopy, fluorescence, hemolymph, microplastic, in vivo

## Abstract

Microplastic pollution is a growing environmental issue, but its effects on the immune systems of invertebrates are not well understood. The Black Soldier Fly (*Hermetia illucens*) larva is an important species because it is widely used in waste management and may help clean up pollutants. This study examined how microplastics interact with the larvae’s immune system. We injected fluorescent polystyrene microbeads directly into the larvae’s body cavity, allowing us to study immune reactions without interference from the gut. Samples were analyzed at different time points up to seven days using histology, cytology, and confocal microscopy. The results showed that microplastics spread throughout the larvae’s tissues and persisted for at least a week. The immune cells (hemocytes) were observed actively engulfing the particles, indicating a clear immune response. Microplastics were also found inside important metabolic and detoxification organs, such as the fat body and Malpighian tubules, raising concerns about potential impacts on larval health and metabolism. Overall, the study demonstrates that microplastics are taken up by immune cells, retained in tissues, and may interfere with physiological functions in Black Soldier Fly larvae. These findings are important for assessing the safety of using this species in bioconversion and environmental cleanup efforts.

## 1. Introduction

Microplastic pollution has become a critical environmental crisis, and the Black Soldier Fly Larva (BSFL) has emerged as a promising tool for organic waste management [1] and potential bioremediation [2,3]. Existing research has shown that BSFL can ingest microplastic particles [4], but this process is limited by the size of the particles, typically to those smaller than the larvae’s mouth opening [5,6]. While larvae appear unable to fully degrade robust plastics like polyethylene, their gut microbiota are known to play a role in the biotransformation of certain polymers and associated plasticizers [7]. Studies also indicate that bioaccumulation factors for these compounds remain low [8], suggesting the larvae efficiently metabolize or eliminate them [5]. Despite this, there is some conflicting evidence regarding the impact of microplastics on larval growth, with some reports suggesting potential disruptions [2,5,6,9]. The BSFL possesses a central digestive tract specialized for nutrient absorption [10,11], surrounded by a metabolically active fat body responsible for energy storage and physiological regulation [12,13,14]. Excretion and osmoregulation are mediated by Malpighian tubules branching from the posterior gut [15]; Refs. [15,16], which filter hemolymph in association with the open circulatory system [17]. This anatomical organization places hemolymph in direct contact with major internal tissues, facilitating the transport of nutrients, metabolites, and potential contaminants [18].

The immune system of *H. illucens* larvae is a sophisticated and rapidly activated innate mechanism composed of both cellular and humoral components [19]. The cellular response, driven by circulating hemocytes, involves phagocytosis and encapsulation, while the humoral response includes the phenoloxidase system and the production of antimicrobial peptides [19,20,21,22]. The kinetics of these responses have been well-characterized through studies involving the injection of foreign bodies, such as bacteria and charged beads [19]. These studies demonstrate that cellular defenses are activated rapidly [23,24], while humoral components follow later, with the specific response being highly dependent on the nature of the foreign body.

While previous research has focused on the ingestion and digestive fate of microplastics, the systemic immune response to these particles when they bypass the gut and enter the body cavity remains completely unexplored. Our study directly investigates the larvae’s immune response by injecting fluorescent microplastic particles into the hemocoel. This new approach enables us to quantify and observe the immune challenge directly, eliminating variables introduced by digestion. The aim of the study is to determine the cellular reactions to microplastic particles and track their movement and accumulation within larval tissues at different developmental stages. This research will provide insights into the immunotoxicological effects of microplastics on BSFL regarding their safe and effective use in bioremediation.

## 2. Materials and Methods

### 2.1. Husbandry and Selection

The study was conducted on 12 BSFL, divided into five experimental groups (*n* = 2 per group) and one control group. As no established protocol was available for injecting fluorescent microplastic particles into *H. illucens* larvae, this pilot study employed two larvae per group to validate husbandry conditions and experimental methods prior to scale-up. This approach minimized risk given the uncertainty in expected outcomes and ensured methodological feasibility before increasing sample size in subsequent experiments. The larvae were housed in plastic cages under controlled environmental conditions, maintaining an ambient temperature of 22–23 °C and relative humidity of 55% [25]. They were provided with unrestricted access to standard chicken feed in moisturized pellet form, ensuring optimal nutritional support for growth and development. BSFL were sourced from Nasekomo EAD in Lozen, Bulgaria, at the first-instar larvae developmental stage, averaging 0.5 mm in size. The subjects were reared in controlled conditions on moisturized pelleted chicken feed substrate until the fourth-fifth instar level.

### 2.2. Experimental Design

The experimental protocol was designed to assess the persistence of microplastics in BSFL. Ten fourth instar larvae were subjected to a thorough cleaning process. Initially, they were washed with tap water to remove any residual dietary debris. After cleaning, the larvae were anesthetized by placing them on a cold plate (ice) for 10 s (Figure 1). Using a Hamilton 700 × 10 µL syringe (Hamilton, Reno, NV, USA), a 5 µL aqueous suspension of carboxylate-modified polystyrene latex beads (fluorescent yellow-green, mean size 2.0 µm, 25 µg/µL, excitation ≈ 470 nm, emission ≈ 505 nm, Sigma Aldrich L4530-1ML, St. Louis, MO, USA) was injected into each larva between the third and penultimate metamere, as described in a previously published article [19]. Larvae were sacrificed at specific time intervals: 1 h, 6 h, 24 h, 48 h, and 7 days post-injection. Hemolymph from one larva per group was collected, stained using the Diff-Quik method [26], and examined under both optical and confocal microscopy (CSLM). The same larvae used for hemolymph collection were injected with 10% formaldehyde (0.5 µL volume) for subsequent histological analysis. Another larva was collected on ice for cryosectioning and CSLM. Larvae sacrificed at 48 h and 7 days were weighed before microplastic administration and post-sacrifice to monitor any changes in body mass. In the same manner, the control group was weighed on the first and last days of the experiment (Figure 1 and Table 1).

### 2.3. Histopathological Analysis

Larvae fixed in 10% formalin were processed for histological examination. Samples were dehydrated in graded ethanol baths and clarified in xylene before embedding. Thin sections (2 µm) were prepared using a rotary microtome. Sections were deparaffinized in xylene and rehydrated through descending ethanol concentrations. Tissue sections were stained using an H&E Staining Kit (ab245880) from Abcam (Cambridge, MA, USA). Hematoxylin was applied to visualize nuclear structures, followed by eosin for cytoplasmic staining. Dehydrated sections were mounted with Permount and coverslips. Slides were analyzed using an Olympus BX51 microscope (Olympus, Tokyo, Japan). Bright-field images were captured with an Olympus SP350 digital camera (Olympus, Tokyo, Japan) and processed using Olympus cellSens software for detailed analysis.

### 2.4. Confocal Scanning Laser Microscopy

Fluorescent confocal imaging was performed using a Zeiss LSM 710 confocal laser scanning microscope (Zeiss, Oberkochen, Germany) using Zen Black Edition 2.1 software, mounted on an Axio Observer Z1 inverted microscope (Zeiss, Oberkochen, Germany). Cellular nuclei were visualized with the fluorescent dye Draq5 (DRAQ5^®^, Cat. No. 4084, Cell Signaling Technology, Danvers, MA, USA), using an excitation wavelength of 543 nm and an emission bandwidth of 661–759 nm. All images were collected under standardized and fixed acquisition settings to enable direct comparison across samples, including a 30 s acquisition time, 20% laser power, 56 µm pinhole diameter, and scan zoom of 1.0, producing 512 × 512-pixel images over a 135 × 135 µm field of view.

### 2.5. Microplastic Assessment

Microplastic localization within larval tissues and hemolymph was analyzed using CSLM. Larvae collected on ice were sectioned and imaged under standardized conditions to ensure consistency in data acquisition. Subsequently, the samples were embedded in OCT (Optimal Cutting Temperature) medium (Poly Freeze, Sigma-Aldrich, Darmstadt, Germany) and frozen using dry ice. Serial cryosections (20 μm thick) were prepared at −20 °C using a cryostat (CM 1850, Leica, Nussloch, Germany) and placed on poly-l-lysine-coated glass slides. After rinsing with distilled water, the nuclei were stained for 5 min with a 1:5000 dilution of blue pseudocolor according to the manufacturer’s instructions. Coverslips were mounted with an aqueous antifade reagent (Mowiol 4-88, Carl Roth, Karlsruhe, Germany), and the slides were examined immediately.

## 3. Results

### 3.1. Normal Anatomy and Histology of BSFL

The presence of fluorescently labeled particles demonstrates that ingested plastics are not confined to the gut lumen but can cross epithelial barriers and circulate systemically. Their distribution in the hemolymph implies potential interactions with the fat body and excretory pathways, raising questions about how such particles may influence metabolism, immunity, and detoxification in insect larvae. Thus, the combination of anatomical knowledge and histological visualization provides essential context for interpreting the biological implications of microplastic uptake and circulation in invertebrate systems (Figure 2).

### 3.2. Weight Variation Before and After Administration

Before microplastic administration, all larvae were weighed, measuring between 0.0102 g and 0.0976 g. Before sacrifice, the larvae of experimental groups 4 and 5 and the control group were again weighed (Table 2).

The analysis of larval body weights before and after microplastic exposure indicates heterogeneous growth patterns in *Hermetia illucens*. Baseline weights displayed considerable variability (0.0102–0.0976 g), reflecting natural differences in larval development. In groups subjected to early sacrifice (≤24 h), only initial measurements were available. After 48 h of exposure, larvae exhibited slight decreases in body mass, potentially linked to handling stress or acute physiological responses to microplastics. However, the absence of a 48 h control group makes it difficult to draw definitive conclusions. Conversely, at 7 days, larvae demonstrated marked weight increases, in some cases exceeding a twofold gain, indicating that growth trajectories were largely maintained despite microplastic administration. Control larvae similarly showed robust growth, consistent with expected developmental dynamics, although one individual displayed an anomalously high increase, likely due to measurement error or exceptional growth capacity (Figure 3).

### 3.3. Microplastic Confocal Scanning Laser Microscopy Analysis

Fluorescence microscopy confirmed the presence of ingested microplastics within the hemolymph of BSFL. Particles were clearly distinguishable by their strong green fluorescence signal, which contrasted with the diffuse red autofluorescence background of the hemolymph. Intensity profile analyses across regions of interest revealed sharp, localized peaks in the green channel, corresponding to individual microplastic particles, while the red channel showed only baseline background variation without defined peaks. The ability to spectrally separate particle fluorescence from hemolymph signals provides a reliable method for confirming microplastic uptake and systemic distribution in insect larvae (Figure 4).

### 3.4. Hemolymph Analysis

Cytological analyses further demonstrated the interaction of hemocytes with ingested microplastics. Light microscopy revealed hemocytes containing intracellular inclusions, consistent with the phagocytic uptake of foreign particles. CSLM provided higher-resolution confirmation, showing distinct fluorescent signals corresponding to microplastic aggregates within hemocytes. The green fluorescence of the particles was clearly localized inside the cellular boundaries, verifying internalization rather than external adherence. (Figure 5).

### 3.5. Histological Analysis

The gut epithelium maintained its integrity, with no signs of epithelial degeneration, necrosis, or sloughing. The muscles and cuticle preserved their typical morphology and organization, with no disruptions or lesions. The fat body showed a normal vacuolated appearance, with no cytoplasmic shrinkage, abnormal accumulation, or lytic changes. Importantly, no indications of toxic effects, inflammatory infiltrates, cellular damage, or tissue disorganization were observed in any of the evaluated sections. These findings confirm the absence of histopathological changes and support that the experimental conditions did not induce detrimental effects in larval tissues (Figure 6).

### 3.6. Confocal Scanning Laser Microscopy Analysis

In larval tissue samples exposed to FITC-labeled microplastics, green fluorescent signals were detected against the nuclear counterstain DRAQ5. No specific fluorescence was observed in the control group, confirming the absence of background signal. After 1 h of exposure, strong FITC fluorescence was evident in association with tissue structures, indicating early adherence and uptake of microplastic particles. At 6 h, microplastic aggregates were present within the tissue microenvironment and closely associated with hemocyte-like cells, suggesting active cellular interaction. By 24 h and 48 h, these aggregates persisted but became more localized, consistent with progressive internalization or sequestration within tissues. Even after 7 days, residual fluorescence signals were still detectable, though at lower intensity, demonstrating long-term retention of microplastics in the larval tissues (Figure 7).

## 4. Discussion

The present study provides novel insights into the immunological and histopathological consequences of systemic microplastic exposure in BSFL. By directly injecting fluorescent carboxylate-modified polystyrene beads into the hemocoel, we bypassed gut-associated barriers and digestion-related variables, allowing us to investigate the immediate and systemic immune responses to microplastics.

To our knowledge, this is the first study to demonstrate the persistence, cellular uptake, and tissue-level distribution of microplastics in BSFL following direct entry into the body cavity. Our findings demonstrate that hemocytes readily phagocytose microplastic particles, consistent with their recognized function in insect innate immunity. In addition to circulating particles through the hemolymph, hemocytes actively sequester them via phagocytosis. This response indicates immune recognition of microplastics as foreign particulates and underscores the role of hemocytes as a frontline defense against circulating contaminants in the insect open circulatory system [19,20]. Although most studies have focused on the ingestion of microplastic particles, our results raise questions about whether non-ingested microplastics can cross the gut barrier and reach the hemolymph through mechanisms such as endocytosis, paracellular transport, or epithelial damage [27,28]. Once in the hemolymph, these particles may interact with immune cells and trigger inflammatory or stress responses [29]. The *Diptera* order, which includes *H. illucens*, comprises numerous species that exhibit stress responses to microplastic exposure, resulting in disrupted metabolic processes, altered weight gain, and changes in developmental timing. Although it remains unclear whether this represents a characteristic of the entire order, microplastics clearly have the capacity to induce pathological responses [30], though one study implies that low doses of microplastics can be beneficial to insects [31]. By directly injecting fluorescent microplastics into the hemocoel, our study specifically examines these internal interactions and immune effects while avoiding the variability associated with gut passage. This approach provides clearer insight into how microplastics can influence internal tissues after translocating beyond the digestive system. The detection of fluorescent particles within hemocytes confirms that microplastics are recognized as foreign bodies and subjected to phagocytic sequestration, similar to responses observed against pathogens or charged beads [9,22]. This cellular interaction suggests that microplastics, despite being chemically inert, are immunologically active stimuli in invertebrate systems.

While the phagocytosis observed here did not result in obvious tissue damage, chronic exposure in natural settings could potentially affect the immune system, demanding further investigation. CSLM revealed that microplastics persisted in larval tissues for at least seven days. The long-term sequestration of particles may represent a detoxification strategy, but it also raises questions about possible interference with nutrient metabolism or osmoregulation. Importantly, no histopathological alterations were detected in the gut, muscle, or fat body, suggesting that acute exposure to microplastics at the tested concentrations does not affect tissue integrity. In the same manner, *Bombyx mori* larvae react similarly to microplastic contamination, with brief changes in the Malpighian tubules and in the hemocytes [32]. Nonetheless, more sensitive markers of oxidative stress, apoptosis, or inflammation may be required to detect subtle subcellular effects not captured by routine H&E staining.

Body weight measurements revealed heterogeneous growth trajectories. While larvae sacrificed at 48 h displayed slight weight reductions, those maintained for 7 days generally showed robust growth, comparable to controls. The temporary weight reduction observed at 48 h post-exposure may reflect short-term physiological costs of microplastic internalization, including immune activation and stress-related metabolic reallocation [33,34]. Organic and fatty acid metabolism, including compounds such as fumaric and stearic acid, appears to be sensitive to microplastic contamination. Recent studies on aquatic insects have highlighted various metabolic abnormalities caused by microplastics [35], which may also help explain the nonuniform larval growth patterns observed in our study. Because intermediate control weights were not recorded, we cannot rule out handling stress or natural metabolic variation as contributing factors. The subsequent recovery and weight gain by day 7 indicate that this response was brief. Future studies should include intermediate control measurements and larger sample sizes to clarify whether this early decrease is microplastic-specific or a general stress effect. These results suggest that systemic microplastic exposure does not impair short-term development, consistent with previous ingestion-based studies reporting minimal or inconsistent effects on BSFL growth [2,5]. However, this contrasts with a recent study that showed that microplastic contamination of the rearing substrate, while not affecting mortality, did cause a delay in larval development [2,3,5].

However, the variability observed here highlights the need for larger sample sizes and longer-term monitoring to determine whether subtle metabolic costs arise from immune engagement and tissue sequestration of microplastics. Given the increasing use of BSFL in organic waste management and as a sustainable protein source, understanding their interactions with microplastics is critical. Our results indicate that larvae are capable of internalizing and retaining microplastics without obvious pathology, which could have implications for both their bioremediation potential and food/feed safety. On one hand, the ability to isolate microplastics may facilitate their removal from waste streams. On the other hand, the persistence of particles within larval tissues suggests a risk of trophic transfer if larvae are used as feed without processing steps that eliminate contaminants. 

Several limitations must be acknowledged. First, the study employed a small sample size (n = 12), limiting the statistical power and generalizability of results. The small sample size likely amplified individual variability, particularly given the broad initial weight range. Accordingly, results should be viewed as preliminary. Future studies should include ≥5–10 individuals per group to improve robustness and ensure reliable assessment of treatment effects. Second, only polystyrene beads of uniform size were tested, whereas environmental microplastics are heterogeneous in shape, size, and chemical composition. Finally, we focused on short-term responses, leaving the long-term consequences of chronic exposure unexplored. Future studies should employ larger cohorts, diverse particle types, and more comprehensive immunological assays, including molecular markers of stress, to fully elucidate the impact of microplastics on BSFL physiology and their potential role in bioremediation.

## 5. Conclusions

This study demonstrates that BSFL mobilize active responses to systemic microplastic exposure. The direct injection of fluorescent polystyrene beads into the hemocoel revealed that hemocytes rapidly internalize microplastics, indicating their recognition as foreign bodies. Microplastics persisted within larval tissues for at least seven days without evidence of clear histopathological alterations. Growth measurements did not provide conclusive evidence of impairment under the present experimental conditions; however, inter-individual variability suggests that subtle physiological costs cannot be excluded. These findings underscore the apparent tolerance of *H. illucens* larvae to acute microplastic exposure while simultaneously raising concerns regarding long-term sequestration and potential trophic transfer. The results contribute to the evaluation of BSFL in waste management, bioconversion, and feed applications, but further investigations with larger sample sizes, diverse particle types, and extended exposure durations are required to more precisely define the implications for larval physiology and ecological safety.

## Figures and Tables

**Figure 1 insects-16-01169-f001:**
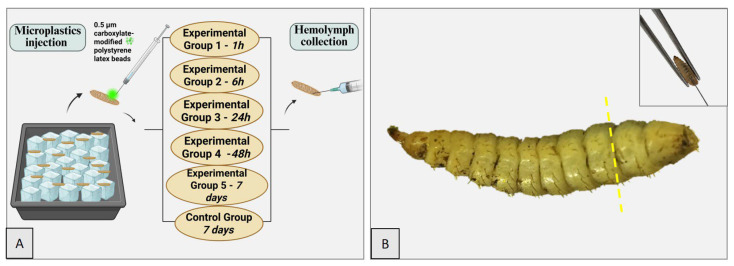
BSFL. Experimental protocol (**A**). Precise site of microplastic administration in the larva, indicated by the yellow dashed line between the third and penultimate metameres (**B**). Inset: Demonstration of the microplastic injection technique using a clamp and a needle.

**Figure 2 insects-16-01169-f002:**
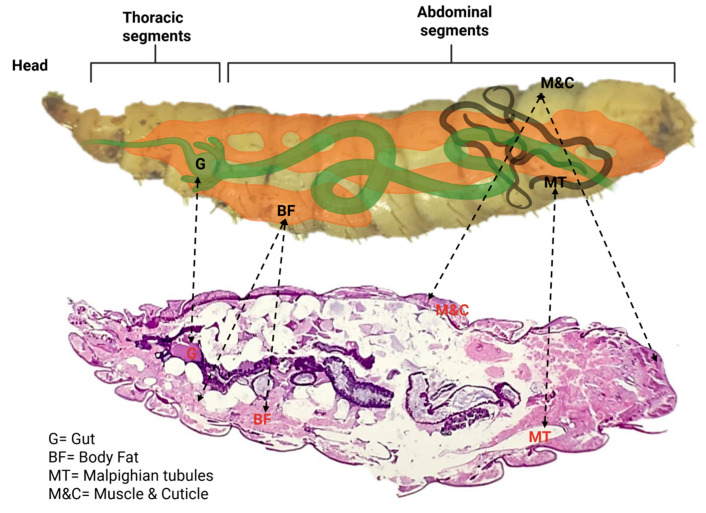
Anatomical organization and histological section of the organ systems of the BSFL. The schematic diagram (**top**) illustrates the major components of the larval organs: the gut (G), body fat (BF), Malpighian tubules (MTs), muscle and cuticle (M&C). The corresponding histological section stained with hematoxylin and eosin (H&E) (**bottom**) highlights these structures in situ within the larval body. The gut (G) is centrally located, surrounded by abundant fat body tissue (BF), while the Malpighian tubules (MTs) extend along the posterior region, reflecting their role in excretion and osmoregulation. Created with BioRender software (https://app.biorender.com/illustrations/68d8f0b3b99fa9878bdc06b8, accessed on 28 September 2025).

**Figure 3 insects-16-01169-f003:**
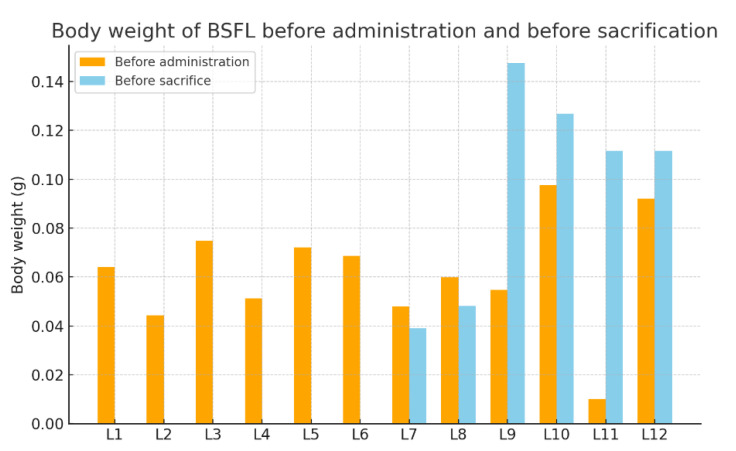
Body weight variations before administration and scarification among experimental groups.

**Figure 4 insects-16-01169-f004:**
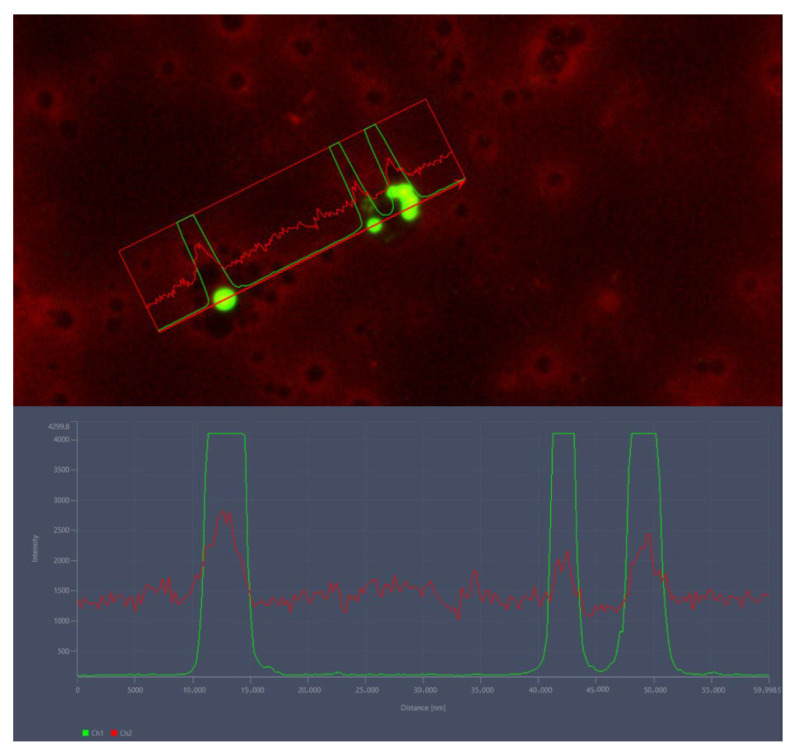
Fluorescent microplastic detection in the hemolymph of BSFL. The above image shows a fluorescence microscopy micrograph with microplastic particles appearing in green (Channel 1, Ch1) against the hemolymph background autofluorescence in red (Channel 2, Ch2). The rectangular region of interest (ROI) highlights a microplastic particle, where fluorescence intensity was measured. The image below presents the fluorescence intensity profile across the ROI. A sharp peak in the green channel (Ch1) corresponds to the localized microplastic particle, while the red channel (Ch2) shows background autofluorescence with no distinct peak.

**Figure 5 insects-16-01169-f005:**
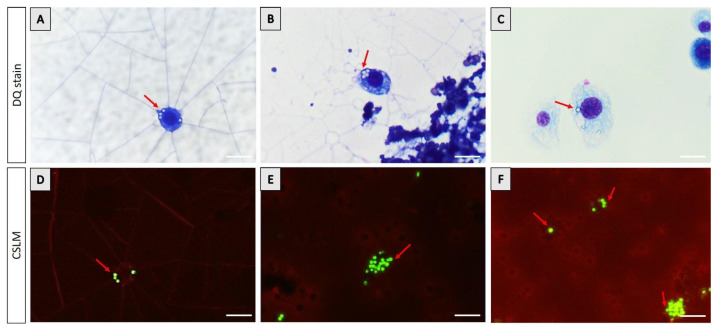
Detection of fluorescent microplastics in the hemolymph of BSFL. (Upper panel, DQ stain) Cytological images showing microplastic ingestion by hemocytes. (**A**–**C**) Hemocytes with visible intracellular inclusions (red arrows) stained in blue, indicating phagocytic uptake of microplastic particles. (Lower panel, CSLM) CSLM images highlighting fluorescent microplastic aggregates (green) internalized by hemocytes (red arrows) within the hemolymph. (**D**–**F**) Distinct fluorescent signals confirm the presence and localization of microplastics inside hemocytes. Scale bars = 10 µm.

**Figure 6 insects-16-01169-f006:**
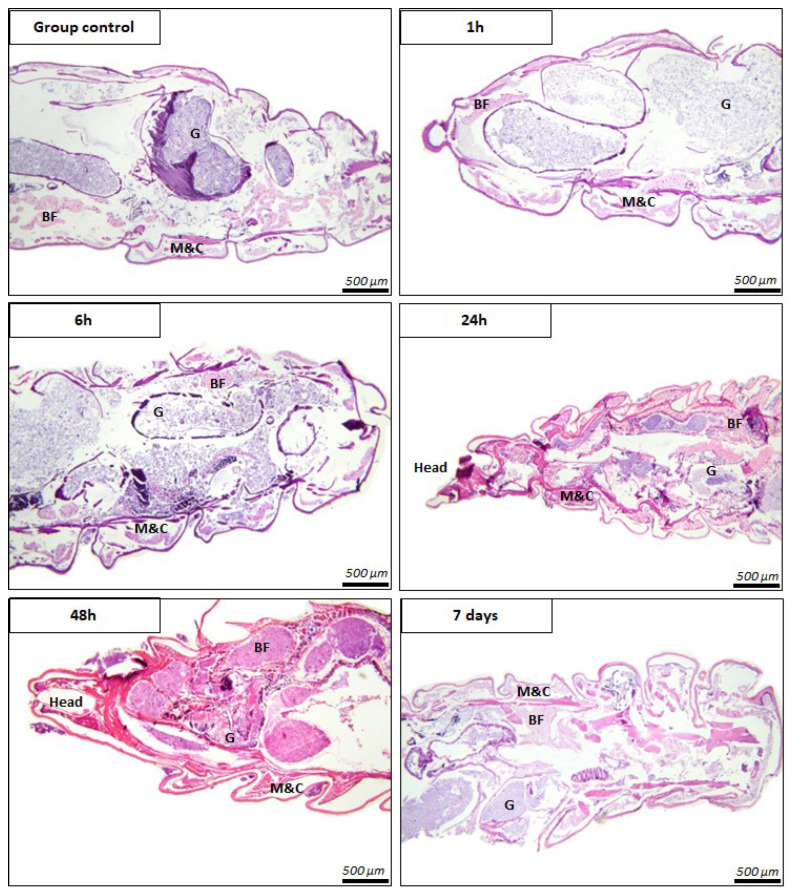
Representative longitudinal sections of larvae from the control group and experimental groups. The main anatomical structures can be identified, including the gut (G), muscle and cuticle (M&C), and fat body (BF). Across all examined time points, the tissues exhibited a normal architecture without evidence of histological alterations. H&E stain, 40× lens.

**Figure 7 insects-16-01169-f007:**
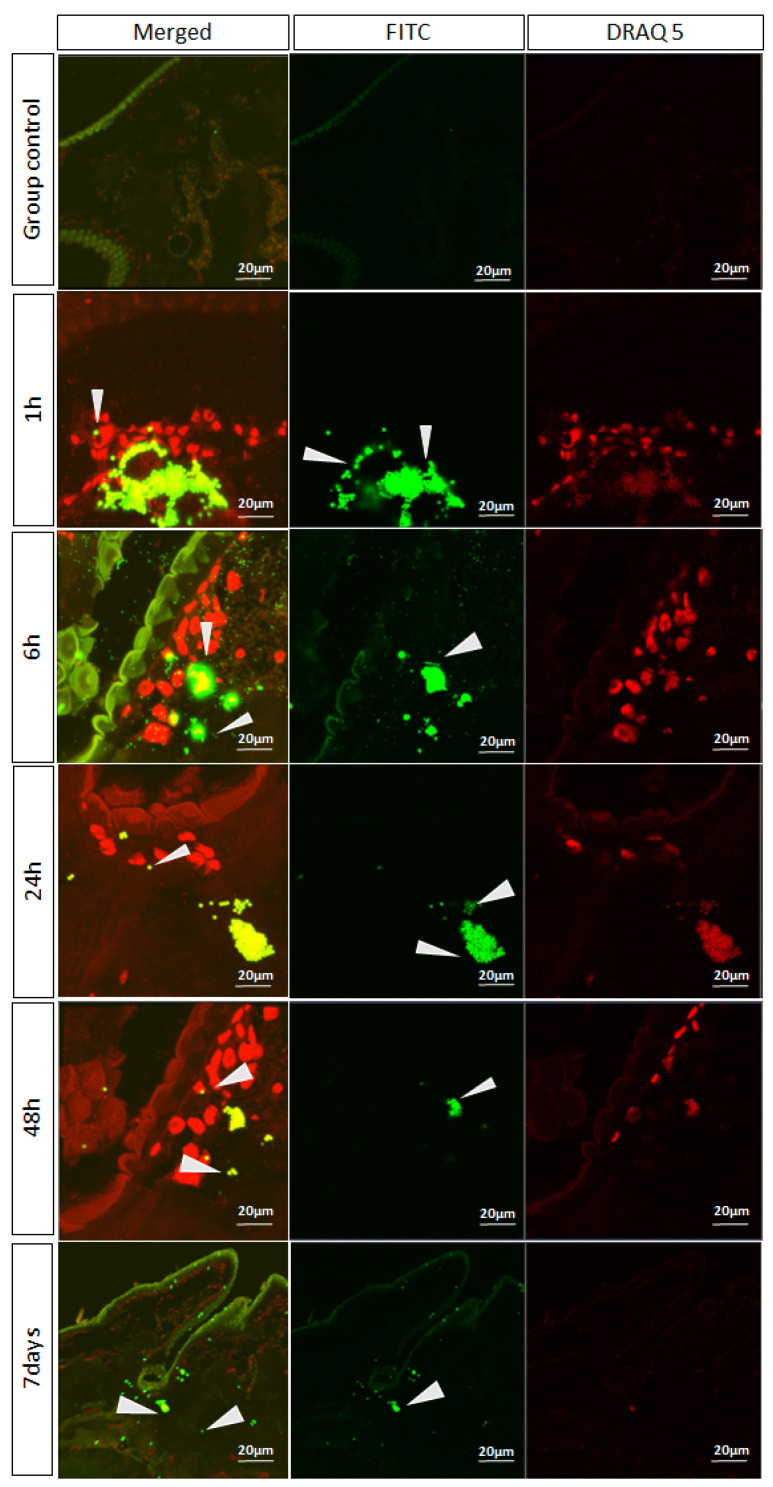
CSLM detection of fluorescent microplastics in BSFL tissues over time. Representative images showing tissue sections exposed to FITC-labeled microplastics (green) and counterstained with DRAQ5 for nuclei (red). Control group tissues displayed no evident fluorescence (top row). At 1 h post-exposure, strong FITC signals (arrowheads) were observed in association with tissue structures, indicating early microplastic adherence and uptake. At 6 h, microplastic aggregates were detected within the tissue microenvironment, closely associated with hemocyte-like cells (arrowheads). By 24 h and 48 h, microplastic clusters persisted but appeared more localized, suggesting progressive cellular internalization or tissue sequestration. After 7 days, residual microplastic fluorescence was still visible (arrowheads), albeit with reduced intensity, indicating long-term retention within larval tissues. Scale bars = 20 µm.

**Table 1 insects-16-01169-t001:** Experimental design.

Groups	Sacrifice	Way of Preservation	Hemolymph Collection
Experimental Group 1	1 h	10% formalin	+
		ice	−
Experimental Group 2	6 h	10% formalin	+
		ice	−
Experimental Group 3	24 h	10% formalin	+
		ice	−
Experimental Group 4	48 h	10% formalin	+
		ice	−
Experimental Group 5	7 days	10% formalin	+
		ice	−
Control Group 6	7 days	10% formalin	+
		ice	−

**Table 2 insects-16-01169-t002:** Body weight measurements.

Groups	Larvae	Body Weight Before Administration	Body Weight Before Sacrifice
**Experimental Group 1**	L1	0.0640 g	-
	L2	0.0444 g	-
**Experimental Group 2**	L3	0.0749 g	-
	L4	0.0513 g	-
**Experimental Group 3**	L5	0.0721 g	-
	L6	0.0687 g	-
**Experimental Group 4**	L7	0.0480 g	0.0390 g
	L8	0.0599 g	0.0482 g
**Experimental Group 5**	L9	0.0547 g	0.1475 g
	L10	0.0976 g	0.1268 g
**Control Group**	L11	0.0102 g	0.1116 g
	L12	0.0920 g	0.1117 g

## Data Availability

The original contributions presented in this study are included in the article. Further inquiries can be directed to the corresponding author.

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
