# Peer review of "Cellular Uptake and Tissue Retention of Microplastics in Black Soldier Fly Larvae"

_insects, 2025, doi:10.3390/insects16111169_

Round 1

Reviewer 1 Report

Comments and Suggestions for Authors

 The study demonstrates that microplastics  are taken up by immune cells, retained in tissues, and may interfere with physiological  functions in Black Soldier Fly larvae. These findings are important for assessing the safety  of using this species in bioconversion and environmental cleanup efforts.

  1. Format Issue: Disorganized Structure of Table 1 (Lines 90–99)   The layout of Table 1 ("Experimental design") is confusing and lacks clarity. For instance, the "Groups" column is not properly aligned with the "Sacreification" time (e.g., "Experimental Group 1" is separated from its corresponding "1h" sacrifice time), and the symbols ("+" or "-") for "Way of preservation" and "Hemolymph collection" are mispositioned. This misalignment makes it difficult for readers to associate each group with its specific treatment and sample processing method.   Recommendation: Restructure the table to ensure each experimental group (e.g., Experimental Group 1–5, Control Group 6) is directly linked to its sacrifice time. Standardize the placement of "+/-" symbols to clearly indicate whether hemolymph was collected or which preservation method was used for each group.
  2. Content Format Error: Incorrect Decimal Separators in Table 2 (Lines 162–177)   Table 2 ("Body weight measurements") uses commas as decimal separators (e.g., "0,0640g", "0,0444g") instead of periods, which violates the standard scientific writing convention (e.g., "0.0640g") and may cause misinterpretation of weight data. Additionally, the weight values for individual larvae (e.g., L1 and L2 in Experimental Group 1) are not clearly indented under their respective groups, leading to confusion about data.   Recommendation: Replace all commas with periods for decimal separation. Reformat the table to nest each larva’s weight data (e.g., L1: 0.0640g) under its corresponding group, ensuring clear visual hierarchy.
  3. Methodological Limitation: Insufficient Sample Size (Lines 44–48)   In Section 2.1 ("Husbandry and selection"), the study only uses 12 larvae total, with n=2 per experimental group. Such a small sample size significantly reduces statistical power, making it impossible to account for individual variability (e.g., the large baseline weight range of 0.0102–0.0976g noted in Section 3.2) and limiting the generalizability of results. The Discussion (Lines 305–310) acknowledges this limitation but provides no rationale for using n=2 in the Methods.   Recommendation: Add a brief justification in Section 2.1 for the small sample size (e.g., "This preliminary study used n=2 per group to validate experimental protocols before scaling up"). In the Discussion, emphasize how the small sample size may have impacted the interpretation of weight variations (e.g., Section 3.2) and strongly recommend increasing n to ≥5–10 per group in future studies.
  4. Missing Methodological Detail: Unreported Microplastic Suspension Concentration (Lines 55–60)   Section 2.2 ("Experimental Design") describes preparing a "0.5 µl aqueous suspension of carboxylate-modified polystyrene latex beads" and injecting 5 µl per larva, but the concentration of the microbeads (e.g., number of beads/µl or µg beads/µl) is not specified. Without this critical information, other researchers cannot replicate the exact microplastic exposure dose, compromising the study’s reproducibility.   Recommendation: Supplement Section 2.2 with the concentration of the carboxylate-modified polystyrene latex bead suspension (e.g., "The suspension contained 1×10⁶ beads/µl" or "5 µg/µl") and reference the Sigma Aldrich product’s specifications (e.g., catalog number) to confirm bead density.
  5. Logical Inconsistency: Unaddressed Discrepancy in Weight Variation (Lines 178–188)   Section 3.2 ("Weight variation before and after administration") notes that Experimental Group 5 (7 days post-injection) showed "marked weight increases" (e.g., L9: 0.0547g to 0.1475g), while the control group also had robust growth. However, the authors do not explain why Experimental Group 4 (48 hours) exhibited "slight decreases in body mass"—they only speculate it may be due to "handling stress or acute physiological responses" without linking this to microplastic exposure (e.g., no comparison to control group’s 48-hour weight, which is unreported).   Recommendation: Add weight data for the control group at 48 hours to clarify whether the weight loss in Experimental Group 4 is microplastic-induced or due to general stress. In the Discussion, elaborate on potential mechanisms (e.g., immune activation energy cost) that could explain the 48-hour weight decrease, and discuss why this effect was not observed at 7 days.

Author Response

Reviewer Comment 1: “Format Issue: Disorganized Structure of Table 1 (Lines 90–99)   The layout of Table 1 ("Experimental design") is confusing and lacks clarity. For instance, the "Groups" column is not properly aligned with the "Sacreification" time (e.g., "Experimental Group 1" is separated from its corresponding "1h" sacrifice time), and the symbols ("+" or "-") for "Way of preservation" and "Hemolymph collection" are mispositioned. This misalignment makes it difficult for readers to associate each group with its specific treatment and sample processing method.   Recommendation: Restructure the table to ensure each experimental group (e.g., Experimental Group 1–5, Control Group 6) is directly linked to its sacrifice time. Standardize the placement of "+/-" symbols to clearly indicate whether hemolymph was collected or which preservation method was used for each group..”

Response:

We thank the reviewer for pointing out the formatting issues in Table 1. In the revised manuscript, Table 1 (“Experimental design”) has been completely reformatted to improve clarity and readability. Each experimental group is now directly aligned with its corresponding sacrifice time, and the “+ / –” symbols for “Way of preservation” and “Hemolymph collection” have been standardized and properly aligned within their respective columns. The revised layout presents the information in a much organized manner, allowing readers to easily associate each group with its specific treatment and sampling conditions.

Reviewer Comment 2: “Content Format Error: Incorrect Decimal Separators in Table 2 (Lines 162–177)   Table 2 ("Body weight measurements") uses commas as decimal separators (e.g., "0,0640g", "0,0444g") instead of periods, which violates the standard scientific writing convention (e.g., "0.0640g") and may cause misinterpretation of weight data. Additionally, the weight values for individual larvae (e.g., L1 and L2 in Experimental Group 1) are not clearly indented under their respective groups, leading to confusion about data.   Recommendation: Replace all commas with periods for decimal separation. Reformat the table to nest each larva’s weight data (e.g., L1: 0.0640g) under its corresponding group, ensuring clear visual hierarchy..”

Response:

We agree with the reviewer’s observation. We modified Table 2 accordingly.

Reviewer Comment 3: “Methodological Limitation: Insufficient Sample Size (Lines 44–48)   In Section 2.1 ("Husbandry and selection"), the study only uses 12 larvae total, with n=2 per experimental group. Such a small sample size significantly reduces statistical power, making it impossible to account for individual variability (e.g., the large baseline weight range of 0.0102–0.0976g noted in Section 3.2) and limiting the generalizability of results. The Discussion (Lines 305–310) acknowledges this limitation but provides no rationale for using n=2 in the Methods.   Recommendation: Add a brief justification in Section 2.1 for the small sample size (e.g., "This preliminary study used n=2 per group to validate experimental protocols before scaling up"). In the Discussion, emphasize how the small sample size may have impacted the interpretation of weight variations (e.g., Section 3.2) and strongly recommend increasing n to ≥5–10 per group in future studies.”

”Response:

We appreciate this important point. We modified the underlined sections accordingly. Please check the Lines 345-349 and Lines 93-98.

Reviewer Comment 4: “Missing Methodological Detail: Unreported Microplastic Suspension Concentration (Lines 55–60)   Section 2.2 ("Experimental Design") describes preparing a "0.5 µl aqueous suspension of carboxylate-modified polystyrene latex beads" and injecting 5 µl per larva, but the concentration of the microbeads (e.g., number of beads/µl or µg beads/µl) is not specified. Without this critical information, other researchers cannot replicate the exact microplastic exposure dose, compromising the study’s reproducibility.   Recommendation: Supplement Section 2.2 with the concentration of the carboxylate-modified polystyrene latex bead suspension (e.g., "The suspension contained 1×10⁶ beads/µl" or "5 µg/µl") and reference the Sigma Aldrich product’s specifications (e.g., catalog number) to confirm bead density..
Response: We appreciate this important point. We added the requested informations. Please check the Lines 104-108.Using a Hamilton 700 10-µL syringe (Hamilton, USA) a 5 µl aqueous suspension of carboxylate-modified polystyrene latex beads (fluorescent yellow-green, mean size 2.0 µm, 25µg/µl Sigma Aldrich L4530-1ML) was injected into each larva between the third and penultimate metamere

Reviewer Comment 5: “Logical Inconsistency: Unaddressed Discrepancy in Weight Variation (Lines 178–188)   Section 3.2 ("Weight variation before and after administration") notes that Experimental Group 5 (7 days post-injection) showed "marked weight increases" (e.g., L9: 0.0547g to 0.1475g), while the control group also had robust growth. However, the authors do not explain why Experimental Group 4 (48 hours) exhibited "slight decreases in body mass"—they only speculate it may be due to "handling stress or acute physiological responses" without linking this to microplastic exposure (e.g., no comparison to control group’s 48-hour weight, which is unreported).   Recommendation: Add weight data for the control group at 48 hours to clarify whether the weight loss in Experimental Group 4 is microplastic-induced or due to general stress. In the Discussion, elaborate on potential mechanisms (e.g., immune activation energy cost) that could explain the 48-hour weight decrease, and discuss why this effect was not observed at 7 days.”

Response: We thank the reviewer for this valuable observation. Intermediate weight measurements at 48 hours were recorded only for exposed larvae in this preliminary study, as the primary aim was to characterize short-term responses to microplastic injection. Control larvae were measured at baseline and endpoint only; future experiments will include parallel intermediate measurements for control groups. The brief reduction in body weight observed in the 48-hour post-exposure group may reflect short-term physiological costs associated with microplastic internalization, including immune activation, stress-response signaling, and temporary metabolic reallocation away from growth. Because intermediate weight measurements were not collected for the control group, we cannot definitively exclude the contribution of handling-related stress or baseline metabolic fluctuations during this early phase. Nevertheless, the recovery and subsequent weight increase observed by day 7 suggest that this response was brief. Future studies incorporating intermediate control measurements and larger sample sizes will help clarify whether the 48-hour weight decrease represents a microplastic-specific acute effect or a general stress response. Also please check the Lines 318-321

Reviewer 2 Report

Comments and Suggestions for Authors

This study investigates the effects of systemic microplastic exposure on the immunity and tissue pathology of Hermetia illucens larvae. However, I still have the following concerns, and I recommend reconsideration after revision.

1.The number of references cited in the Introduction section is relatively limited.

2.The Introduction or Discussion should include a description of how ingested microplastics can translocate from the digestive tract to other tissues such as the hemolymph, and how this process affects the immune system. This would help highlight the advantages of studying microplastics that directly enter the hemolymph.

3.The sample size appears too small, with only two larvae per group. Some parameters (e.g., those shown in Figure 3) lack sufficient replication for statistical analysis.

4.Line 105: “µl” should be written as “µL.” Please carefully check all formatting details throughout the manuscript.

5.Figure 1A: The text is too small, especially the labels above the syringe.

6.Line 90: The spacing around the temperature symbol is inconsistent — please standardize.

7.The Discussion should be divided into paragraphs; currently, it is written as a single block.

8.In Figure 6, each treatment time point should have its own untreated control, rather than sharing a single common control.

9.The Discussion section includes few comparisons with related studies and repeats several references already cited in the Introduction.

Author Response

Reviewer Comment 1: “The number of references cited in the Introduction section is relatively limited.”

Response:We thank the reviewer for pointing out this issue. We acted accordingly, adding supplimenary references. Please check the Introduction for: (Wang & Shelomi, 2017), (Nsiah-Gyambibi et al., 2025), (Dam et al., 2024), (Liu et al., 2024), (Shah et al., 2025)

Reviewer Comment 2: “The Introduction or Discussion should include a description of how ingested microplastics can translocate from the digestive tract to other tissues such as the hemolymph, and how this process affects the immune system. This would help highlight the advantages of studying microplastics that directly enter the hemolymph.”

Response:We appreciate this insightful suggestion. In the revised Discussion, we have now included a paragraph describing the potential translocation of ingested microplastics from the digestive tract to other tissues such as the hemolymph, and the associated immune implications. Specifically, we discuss how microplastics can cross the intestinal barrier through mechanisms such as endocytosis, paracellular transport, or physical damage to epithelial tissues, allowing them to enter the circulatory system. Once in the hemolymph, these particles may interact directly with immune cells, potentially triggering inflammatory or stress responses. This addition emphasizes the relevance of investigating microplastics that directly enter the hemolymph, as it provides insight into the systemic effects beyond initial ingestion. Please check Lines 284-289

Reviewer Comment 3: “The sample size appears too small, with only two larvae per group. Some parameters (e.g., those shown in Figure 3) lack sufficient replication for statistical analysis.”

Response:We appreciate this important point. We added explanatory information regarding the decision of using a small sample size and current limitations. Please check the Lines 93-98 and 345-349

Reviewer Comment 4: “Line 105: “µl” should be written as “µL.” Please carefully check all formatting details throughout the manuscript

Response: We appreciate this important point. We modified the requested sequence

Reviewer Comment 5: “Figure 1A: The text is too small, especially the labels above the syringe..”

Response: We thank the reviewer for this valuable observation. We modified the Figure 1A accordingly.

Reviewer Comment 6: “Line 90: The spacing around the temperature symbol is inconsistent — please standardize.”

Response: We thank the reviewer for this valuable observation. We modified the structures accordingly. All spaces regarding the temperature symbol are standardised.

Reviewer Comment 7: “The Discussion should be divided into paragraphs; currently, it is written as a single block.

Response: We thank the reviewer for this valuable observation. We modified the “Discussion” section in several paragraphs. Feel free to check the “Discussion” section.

Reviewer Comment 8: “In Figure 6, each treatment time point should have its own untreated control, rather than sharing a single common control.”

Response: We thank the reviewer for this valuable observation. In this study, we conducted experiments on a single experimental group, and therefore, we do not have multiple untreated controls corresponding to each treatment time point. Generating 5–6 separate experimental groups was not feasible within the scope of this work. However, if the reviewer requests, we can provide additional histological images from the same experimental group to further illustrate the findings at different time points.

Reviewer Comment 9: “The Discussion section includes few comparisons with related studies and repeats several references already cited in the Introduction.

Response: We thank the reviewer for this insightful comment. We have revised the Discussion section to emphasize the relevance of using a direct injection model, the impact of microplastic toxicity across different larval tissues, and to provide a more thorough comparison with previous studies, highlighting both consistent and contradictory findings. Additionally, the list of references has been updated to avoid redundancy and to better support the discussion.

Reviewer 3 Report

Comments and Suggestions for Authors

The authors submitted a manuscript that describes a very detailed study, in which black soldier fly larvae were exposed to microplastic beads by injection and various subsequent analyses were performed. Overall, the manuscript reads well, is correctly structured, and was interesting to read. I have some comments which I believe should be addressed. Most of these are of a minor nature, and are provided below.

More importantly, I have serious concerns about the robustness of the in vivo exposure parts and the conclusions drawn from it. If I understand correctly, only n=2 individual larvae were tested. I understand that the resource intensity of the various analyses necessitate some limitations in the experimental design. I appreciate that these are recognized and discussed as such, but this doesn't justify the choices fully. I believe that these results (and related discussion) should either be largely omitted, with focus lying almost exclusively on the analyses; or that an additional experiment should be ran in which larger groups of larvae are exposed via their diet - although I understand that this will have its own downsides. Due to the nature of BSFL to aggregate, any experimental design that does not permit this behavior will influence the results. Various aspects related to outliers, etc. should similarly be addressed. 

Some more specific minor comments:

-Abstract: please check the dashes interrupting words (-)

-General: please italicize all species names

-General: please be consistent in using abbreviations and/or scientific or unabbreviated name, e.g. for BSFL/Hermetia illucens as well as CSLM (sometimes written out fully again)

General: inconsistent naming of groups (experiment group 1 / 2h, etc.)

Line 91: please provide a reference related to optimal support for growth. It is suggested that dry pellets are provided as feed, this does not match my understanding of BSFL that require wet substrate, please clarify. Please justify ad libitum feeding, since this may induce fungal growth in certain conditions.

Line 103: what is diff-quick? Assuming it is a standardized protocol (similar to QuEChERS for extraction), but at least a reference to a full description of the protocol is needed

Line 107: why weighing only after 48h and 7 days?

Line 110: the fact that the control group was only weighed after 7 days is not really apparent from the text, only the figure. Why no control for all experimental groups?

Line 115: table adds very little compared to text. The fact that literally all treatments were preserved in the same manner makes it superfluous

Line 118: what was the standard deviation for this group? It can be rather heterogenous in my experience.

Line 134: please name CLSM software and explain the 'standardized settings'

Line 137: please move supplier of DRAQ5 from line 150 to its first mention here

Line 156-168: new theory is introduced in results. Why not in introduction, or split up between intro and discussion? This seems unusual.

Line 195: Lack of a control group for 48H does not support the conclusion that exposure reduces growth

Line 231: leaning towards inclusion in discussion instead. This goes for several other instances where results are directly discussed.

Line 270: lack of nuclei shown in control and 7 days images. Again, would be useful to have controls for all treatments so that staining can be tested between microplastics and background 

figure 7: how were red and green images merged? Assuming a capability of the software, but since this is not mentioned (see comment related to line 134), no way to be sure.

Author Response

Reviewer Comment 1: “More importantly, I have serious concerns about the robustness of the in vivo exposure parts and the conclusions drawn from it. If I understand correctly, only n=2 individual larvae were tested. I understand that the resource intensity of the various analyses necessitate some limitations in the experimental design. I appreciate that these are recognized and discussed as such, but this doesn't justify the choices fully. I believe that these results (and related discussion) should either be largely omitted, with focus lying almost exclusively on the analyses; or that an additional experiment should be ran in which larger groups of larvae are exposed via their diet - although I understand that this will have its own downsides. Due to the nature of BSFL to aggregate, any experimental design that does not permit this behavior will influence the results. Various aspects related to outliers, etc. should similarly be addressed. .”

Response: We thank the reviewer for highlighting this important point and fully recognize the limitation associated with the small sample size in our in vivo exposure experiments. The decision to use only n=2 individual larvae was driven by the absence of previously established experimental protocols for the injection of fluorescently labeled plastic beads, as well as the inherent risk that this novel approach might not yield measurable results. We acknowledge that this sample size significantly limits the statistical power of our conclusions. We would like to respectfully emphasize that the primary goal of these experiments was to serve as a pilot study, aimed at demonstrating feasibility and guiding the design of future, more extensive studies. We have carefully discussed these limitations in the manuscript and would appreciate if the reviewer could consider the exploratory nature of these experiments when evaluating the conclusions drawn. Also please check the Lines 93-97; 345-350

Reviewer Comment 2: “-Abstract: please check the dashes interrupting words (-)”

Response:We appreciate this insightful suggestion. We’ve checked the abstract and removed all the dashes.

Reviewer Comment 3: “General: please italicize all species names.”

Response:We appreciate this important point. We acted accordingly.

Reviewer Comment 4: “General: please be consistent in using abbreviations and/or scientific or unabbreviated name, e.g. for BSFL/Hermetia illucens as well as CSLM (sometimes written out fully again)”

Response: We appreciate this important point. We acted accordingly

Reviewer Comment 5: “General: inconsistent naming of groups (experiment group 1 / 2h, etc.)”

Response: We thank the reviewer for this valuable observation. We chose a more consistent naming of experimental groups. Please check the Lines 189-194

Reviewer Comment 6: “Line 91: please provide a reference related to optimal support for growth. It is suggested that dry pellets are provided as feed, this does not match my understanding of BSFL that require wet substrate, please clarify. Please justify ad libitum feeding, since this may induce fungal growth in certain conditions.”

Response: We thank the reviewer for this valuable observation. We apologize for the misunderstanding regarding the feeding substrate. Although the pellets were initially provided in dry form, they were subsequently moistened prior to feeding in order to ensure optimal rearing conditions for the larvae. Regarding the comment on ad libitum feeding, in our experience this approach has not led to fungal development under the controlled conditions applied in our experiments. The term ad libitum was used solely to indicate that the larvae were not restricted to a predetermined, weighed quantity of feed, but rather had continuous access to sufficient food throughout the experiment. In addition, as suggested, we have now included an appropriate reference supporting the optimal substrate conditions for BSFL growth in the revised manuscript Lines 100-101.

Reviewer Comment 7: “Line 103: what is diff-quick? Assuming it is a standardized protocol (similar to QuEChERS for extraction), but at least a reference to a full description of the protocol is needed”.

Response: We thank the reviewer for this valuable observation. We added the requested reference. Please check Lines 116-118

Reviewer Comment 8: “Line 107: why weighing only after 48h and 7 days?”

Response: We thank the reviewer for this pertinent observation. The authors selected the 48-hour and 7-day time points because it was considered that larvae exposed to the experimental conditions for less than 48 hours would not exhibit measurable or significant differences in weight gain ratio. These intervals were therefore chosen to ensure that any physiological or metabolic effects of the treatments could be meaningfully detected. We acknowledge that additional early time points could provide further insights, and we will consider including such measurements in future studies.

Reviewer Comment 9: “Line 110: the fact that the control group was only weighed after 7 days is not really apparent from the text, only the figure. Why no control for all experimental groups?

.

Response: We thank the reviewer for this valuable comment and have now added a clarifying sentence in the revised manuscript (Lines 190-191) to explicitly state that the control group was weighed only after 7 days. Regarding the experimental design, we acknowledge the reviewer’s observation and would like to clarify that separate control groups for each experimental condition were not included due to the same constraints that limited our overall sample size. As previously mentioned, this work was conceived as a pilot study, aiming primarily to test the feasibility of the experimental approach rather than to achieve comprehensive statistical replication. Therefore, a single control group was used to provide a general baseline for comparison. We will consider including such measurements in future studies.

Reviewer Comment 10: “Line 115: table adds very little compared to text. The fact that literally all treatments were preserved in the same manner makes it superfluous ?”

Response: We appreciate the reviewer’s thoughtful observation regarding Table 1. The table was originally included with the intention of providing readers with a clear, concise overview of the experimental design and sampling timeline. We recognize, however, that the preservation conditions are uniform across all treatment groups, which may reduce the perceived added value of presenting this information in table form. We are not entirely certain whether the reviewer is recommending that the table be removed. If the suggestion is to eliminate the table to avoid redundancy with the text, we will gladly do so and instead retain only a clear narrative description of the experimental schedule within the Methods section. Conversely, if the reviewer believes that a visual summary would still be beneficial to readers, we are happy to revise the table to better emphasize the experimental structure (e.g., by including sample size and specific downstream analyses for each time point). We kindly ask the reviewer to clarify their preference, and we will revise accordingly.

Reviewer Comment 11: “Line 118: what was the standard deviation for this group? It can be rather heterogenous in my experience..”

Response:We thank the reviewer for this insightful remark. The standard deviation for the initial larval size was not determined experimentally in our study. Instead, we relied on the size and developmental stage information provided by the supplier. We acknowledge that some degree of heterogeneity in larval size is likely, as also noted by the reviewer, and we recognize that this variability may have contributed to differences observed within experimental groups.

Reviewer Comment 12: “Line 134: please name CLSM software and explain the 'standardized settings.”

Response: We thank the reviewer for this observation. We modified the structure accordingly. Please check Lines 142-143; 145-148

Reviewer Comment 13: “Line 137: please move supplier of DRAQ5 from line 150 to its first mention here.”

Response: We thank the reviewer for this observation. We modified the structure accordingly. Please check Lines 144

Reviewer Comment 14: “Line 156-168: new theory is introduced in results. Why not in introduction, or split up between intro and discussion? This seems unusual.”

Response:We thank the reviewer for this valuable comment. We agree that the anatomical description presented in Lines 156–168 provides background information rather than results. Following the reviewer’s suggestion, we have relocated this content to the Introduction, where it now provides relevant biological context for understanding microplastic distribution in BSFL. The Results section now focuses exclusively on experimental observations. Please check Lines 65-71

Reviewer Comment 15: “Line 195: Lack of a control group for 48H does not support the conclusion that exposure reduces growth.”

Response:We appreciate the reviewer’s comment and agree that the absence of a control group at the 48-h time point limits our ability to definitively attribute the transient reduction in body weight to microplastic exposure. As correctly noted, individual variability cannot be excluded as an explanation. Nevertheless, we reported the slight decrease in weight at 48 h because it was an empirical observation in the experimental cohort and, in our view, warranted transparent presentation. We have revised the text to clarify that this finding should be interpreted cautiously and that no firm conclusion can be drawn regarding causality in the absence of a time-matched control group. Please check Lines 190-191

Reviewer Comment 15: “Line 231: leaning towards inclusion in discussion instead. This goes for several other instances where results are directly discussed.”

Response:We appreciate the observation. We have modified the pointed structureaccordingly. Please check the Lines 279-283

Reviewer Comment 16: “Line 270: lack of nuclei shown in control and 7 days images. Again, would be useful to have controls for all treatments so that staining can be tested between microplastics and background

Response:We thank the reviewer for this valuable observation. We acknowledge that the absence of control images for each experimental time point limits the ability to directly compare nuclear staining between microplastic-exposed samples and background tissue. Owing to the pilot nature of this study and the limited number of specimens available, it was not feasible to include a full set of time-matched controls for each sampling point. As a result, we are unable to provide the complete reference panel suggested. Nevertheless, we recognize the importance of this control structure for validating staining consistency, and we agree that future work should incorporate parallel control groups at all time points to strengthen comparative interpretation of histological and imaging data. We have clarified this limitation in the revised manuscript and will address it in subsequent studies.

Reviewer Comment 17: “figure 7: how were red and green images merged? Assuming a capability of the software, but since this is not mentioned (see comment related to line 134), no way to be sure.

Response:We thank the reviewer for this comment. The red and green fluorescence channels shown in Figure 7 were merged using the native channel overlay function of the confocal acquisition software (ZEN Black Edition 2.1, Carl Zeiss). This software allows visualization of each fluorescence channel independently as well as in merged mode, enabling precise identification of co-localization events.

Reviewer 4 Report

Comments and Suggestions for Authors

Cellular uptake and tissue retention of microplastics in Black soldier fly larvae.

The authors present a good body of work that defines the cellular uptake and tissue retention of Microplastics in BSFL. I find the research of interest and has merit due to the rigor of the experimental set up.

i had concerns regarding the sample size which was addressed by the authors and understand that this will be addressed in subsequent studies.

Recommendation – minor revision.

I have several observations largely regarding English usage and editing.  

Line 21 – Syntax: Replace ‘Research team…’ with  ‘We…’

Line 22 – Syntax: ‘…., allowing the study of immune reactions….’

Line 34 -  out of place Hyphen

Line 35 – out of place Hyphen and non-italicized species name.

Line 36, 45, 47, 48 - out of place Hyphen

Line 50 - non-italicized species name.

Line 66 - non-italicized species name.

Line 87 – You have defined Black soldier fly larvae as BSFL in line 58. Please use the acronym.

Line 90 – Was there a day/night cycle?

Line 95 –Please state what instar was used. ‘Ten (what instar (1 to 5)) Instar larvae….’

Line 103 – Please reference or explain the Diff-Quick method

The section 2.3 is a little redundant and repetitive and can be combined into the respective 2.1 and 2.2 sections respectively ie. regarding BSFL rearing 2.1. And the microplastic section can be combined with 2.2.

Is the Draq5 fluro dye the same in line 137 and 150? If so please put the manufacturers details in the former position rather than the latter.

Line 146 – define OCT acronym when used in the text.

Line 156 – use the acronym

The paragraph at line 169 is in my opinion is interpretation and therefore should be in the discussion.

Line 186 / 187 – add in… ‘…administration of microplastic’ best to say what you administered.

The contrast on figure 4 is not great. One cannot see the CH1 peaks on a blue background.  One may have to change the trace colours – CH1 – white would work.

Line 248 – please replace ‘…without..’ with ‘…with no…’ I feel it is more impactful that  these abnormalities are not happening in the preparations and is representative. Exemplified by the next sentence!

Write out CSLM for the 3.6 heading.

There is no mention that the microplastics were FITC labeled in the M and M

Author Response

Reviewer Comment 1: “Line 21 – Syntax: Replace ‘Research team…’ with  ‘We..”

Response: We thank the reviewer for this observation. We acted accordingly. Please check Line 21

Reviewer Comment 2: “Line 22 – Syntax: ‘…., allowing the study of immune reactions….’”

Response:We appreciate this insightful suggestion. We’ve checked and modified accordingly. Please check Line 22

Reviewer Comment 3: “Line 34 -  out of place Hyphen.”

Response:We appreciate this important point. We acted accordingly. Please check Line 34

Reviewer Comment 4: “Line 35 – out of place Hyphen and non-italicized species name.”

Response: We appreciate this important point. We acted accordingly. Please check Line 35

Reviewer Comment 5: “Line 36, 45, 47, 48 - out of place Hyphen”

Response: We thank the reviewer for this valuable observation. We acted accordingly. Please check the Lines 36;45;47-48

Reviewer Comment 6: “Line 50 - non-italicized species name..”

Response: We thank the reviewer for this valuable observation. We modified accordingly.Please check Line 50

Reviewer Comment 7: “Line 66 - non-italicized species name.”.

Response: We thank the reviewer for this valuable observation. We acted accordingly. Please check Line 78

Reviewer Comment 8: “Line 87 – You have defined Black soldier fly larvae as BSFL in line 58. Please use the acronym.”

Response: We thank the reviewer for this pertinent observation. We modified the error accordingly. Please check the Line 100.

Reviewer Comment 9: “Line 90 – Was there a day/night cycle?”

.Response: We thank the reviewer for this question. The larvae were reared under constant environmental conditions without a defined day/night cycle, as light exposure is not considered a critical factor for BSFL development during the larval stage. However, we recognize that photoperiod can influence behavior and metabolism in some cases

Reviewer Comment 10: “Line 95 –Please state what instar was used. ‘Ten (what instar (1 to 5)) Instar larvae….’”

Response: We appreciate the reviewer’s thoughtful observation. We acted accordingly. Please check Line 112

Reviewer Comment 11: “Line 103 – Please reference or explain the Diff-Quick method”

Response:We thank the reviewer for this insightful remark. We added a proper reference. Please check Line 121.

Reviewer Comment 12: “The section 2.3 is a little redundant and repetitive and can be combined into the respective 2.1 and 2.2 sections respectively ie. regarding BSFL rearing 2.1. And the microplastic section can be combined with 2.2.”

Response: We thank the reviewer for this observation. We modified the structure accordingly. Please check Lines 109-112; 120;140-141;

Reviewer Comment 13: “Is the Draq5 fluro dye the same in line 137 and 150? If so please put the manufacturers details in the former position rather than the latter..”

Response: We thank the reviewer for this observation. We modified the structure accordingly. Please check Line 150

Reviewer Comment 14: “Line 146 – define OCT acronym when used in the text..”

Response:We thank the reviewer for this valuable comment. We acted acordingly. Please check Lines 161

Reviewer Comment 15: “Line 156 – use the acronym.”

Response:We appreciate the reviewer’s comment. We modified accordingly. Please check Lines 68

Reviewer Comment 16: “The paragraph at line 169 is in my opinion is interpretation and therefore should be in the discussion.

Response:We appreciate the observation. We have modified the pointed structure accordingly. Please check the Lines 169-170

Reviewer Comment 17: “Line 186 / 187 – add in… ‘…administration of microplastic’ best to say what you administered.

Response:We thank the reviewer for this comment. We modified the structure accordingly. Please check Line 187.

Reviewer Comment 18: “The contrast on figure 4 is not great. One cannot see the CH1 peaks on a blue background.  One may have to change the trace colours – CH1 – white would work..

Response: We thank the reviewer for this comment. A new image was acquired - we preferred to keep Channel 1 in green for consistency with the other figures. However, the intensity and contrast were changed, and additional elements were removed compared to the initial figure, for a better visualisation of the details

Reviewer Comment 19: “Line 248 – please replace ‘…without..’ with ‘…with no…’ I feel it is more impactful that these abnormalities are not happening in the preparations and is representative. Exemplified by the next sentence!

Response:We thank the reviewer for this comment. We modified the structure accordingly. Please check Line 243.

Reviewer Comment 20: “Write out CSLM for the 3.6 heading.

Response:We thank the reviewer for this comment. We modified the structure accordingly. Please check Line 253.

Reviewer Comment 21: “There is no mention that the microplastics were FITC labeled in the M and M”.

Response:We thank the reviewer for noticing the omission. The microplastic particles used in this study were fluorescent yellow-green, carboxylate-modified polystyrene latex beads (Sigma‑Aldrich, product L4530) which have an excitation wavelength of ~470 nm and emission of ~505 nm. We have now updated the Materials & Methods section to explicitly state that these beads were used and are FITC-equivalent labeled for fluorescence detection. Please check Lines 118-120.

Round 2

Reviewer 1 Report

Comments and Suggestions for Authors

The authors have revised the manuscript according to the requirements and it is now approved for publication.

Reviewer 2 Report

Comments and Suggestions for Authors

no

Reviewer 3 Report

Comments and Suggestions for Authors

The authors have submitted a revised version of the manuscript. Aside from some minor editorial issues which can be addressed in the production process, I believe it is acceptable for publication.